# OpenReview forum: "Safe + Safe = Unsafe? Exploring How Safe Images Can Be Exploited to Jailbreak Large Vision-Language Models"
_NeurIPS.cc/2025/Conference — NeurIPS 2025 poster_

### Official Review · Reviewer_fnZy · 2025-06-05

**Clarity:** 3
**Significance:** 3
**Originality:** 3
**Rating:** 5
**Confidence:** 4

**Summary:**

This paper addresses the problem of jailbreaking by exploiting the inherent capabilities of Large Vision-Language Models (LVLMs), which process both images and text and use reasoning to generate outputs. The authors propose the Safety Snowball Agent (SSA), a novel LLM-based agent designed to jailbreak existing LVLMs. SSA operates in two stages: it first generates harmful images and moderately harmful questions related to a harmful topic to elicit an initial response. This initial output is then used to craft a second prompt that further induces the LVLM to produce harmful content. The authors evaluate their method on seven LVLMs, demonstrating that SSA outperforms existing approaches in jailbreak success rate. An ablation study further reveals that combining both visual and textual inputs is critical for effective jailbreaking.

**Questions:**

What are the main reasons behind the failure cases of SSA? Were there specific categories of prompts or types of content to which LVLMs demonstrated particular robustness?

**Ethical Concerns:**

["NO or VERY MINOR ethics concerns only"]

**Final Justification:**

I have read the author responses and all my concerns are resolved. As my original score is already high, I decided to keep my original score.

**Limitations:**

yes

**Quality:**

3

**Strengths And Weaknesses:**

Strengths
- The authors introduce a novel method for jailbreaking that uses safe inputs to trigger the generation of unsafe content.
- The proposed framework is fully automatic and generalizable.
- Extensive experiments are conducted on seven LVLMs, with comparisons against six baseline methods.
- The paper is well-written and easy to follow.

Weaknesses
- The paper lacks illustrative examples, which limits full comprehension of the proposed framework. For instance, including examples of the generated images or the initial question ($p_f$) would enhance clarity.
- Providing a detailed recipe or guideline for generating the initial question ($p_f$) would be helpful, as this step appears to significantly influence the initial response.
- The captions in the tables are too brief and lack sufficient detail to aid comprehension. It would be beneficial to provide more descriptive captions for each table to enhance clarity and ease of understanding.

---

> ### Author Rebuttal · Authors · 2025-07-31
>
> >Q1: What are the main reasons behind the failure cases of SSA? Were there specific categories of prompts or types of content to which LVLMs demonstrated particular robustness?
>
> **A1:**   Thank you for the question. In our analysis of the failure cases, we found that SSA tends to be less effective when the harmful intent identified by the model contains overly sensitive or explicitly flagged keywords. In such cases, LVLMs often immediately reject the request, even when the input image and prompt appear safe. Examples include phrases like “make a bomb,” “child pornography,” or “how to assassinate a politician.” These categories may trigger strong built-in safety filters regardless of prompt phrasing or context. We will add this analysis to our discussion section for further clarification.
>
>
> >Q2: The paper lacks illustrative examples, which limits full comprehension of the proposed framework. For instance, including examples of the generated images or the initial question () would enhance clarity.
>
> **A2:** Thank you for this suggestion. While we already provide a detailed explanation—including generated images, prompt templates, and step-by-step outputs—in Appendix A.8.6, we acknowledge that this may not be sufficiently emphasized in the main text. To address this, we will enhance our framework diagram by embedding a representative jailbreak image along with its initial-question prompt directly into the figure. Additionally, we will add clear references in Sections 3 and 5 to guide readers to these examples.
>
> >Q3: Providing a detailed recipe or guideline for generating the initial question () would be helpful, as this step appears to significantly influence the initial response.
>
> **A3:** Thank you for the helpful suggestion. We agree that the generation of the initial question plays a crucial role in shaping the model’s response. In our current version, the detailed prompt templates and generation process for the initial question are already included in Appendix A.3. In the revision, we will more clearly highlight this step in the main text (Section 3.3) and explicitly refer readers to the appendix to ensure better visibility and understanding of the process.
>
> >Q4: The captions in the tables are too brief and lack sufficient detail to aid comprehension. It would be beneficial to provide more descriptive captions for each table to enhance clarity and ease of understanding.
>
> **A4:** Thank you for this suggestion. We will expand all table captions to include a concise description of each table’s objective, the dataset or models being compared, definitions of the metrics reported, and a brief summary of the key finding.

---

> > ### Comment · Reviewer_fnZy · 2025-08-02
> >
> > Thank the authors for their response, most of my concerns raised in the review are addressed. Since I already gave high review score, I decide to maintain my score. Thank the authors for their hard work!

---

### Official Review · Reviewer_td69 · 2025-06-28

**Clarity:** 2
**Significance:** 2
**Originality:** 2
**Rating:** 3
**Confidence:** 4

**Summary:**

This work proposes a jailbreaking method for LVLMs and demonstrates that even safe images can be exploited to generate harmful responses. The authors
propose the agent-based jailbreaking framework, which utilizes the snowballing behaviors of LVLMs. The results show that the proposed framework achieves a high jailbreaking success ratio.

**Questions:**

Please see the questions above.

**Ethical Concerns:**

["NO or VERY MINOR ethics concerns only"]

**Final Justification:**

My major concern regarding the contribution of this work and clarity has been resolved, and this work reveals a new way of VLM jailbreaking using safe images. That said, I think many aspects of this work can be much improved to understand the underlying mechanisms (e.g., detailed analysis of safety-related components and a more precise connection to the overexplanation behavior and safety issue, although the authors provided a hypothetical explanation), and the technical aspect of the jailbreaking method was straightforward given previous approaches.

**Limitations:**

Yes

**Paper Formatting Concerns:**

There are no major paper formatting concerns.

**Quality:**

2

**Strengths And Weaknesses:**

## Strengths

S1: The proposed method shows a strong attack success rate, raising concerns in VLM safety.

## Weaknesses

W1: Limited novelty - The key intuition of this work follows the 'crescendo' work (https://arxiv.org/abs/2404.01833, which is not cited in this paper) on LLMs. The main contribution of this work is its expansion to various VLM safety domains, but the analysis and insights for the SSA's behavior depending on the domains are not thoroughly explored.

W2: Regarding the subjectivity of 'not overly harmful prompts' used for the initial query, the authors provide supporting data in Table 12. However, the result is only restricted to the violence category, and important statistical information is missing to show the credibility. For example, how many samples were used to evaluate this data? How can we assure that the same result holds for other categories?

W3: The authors propose an insight that 'universal reasoning abilities' of LVLMs as a fundamental reason for VLM safety issues, but this term is not thoroughly defined throughout the paper. Could you clarify which aspect of behaviors you mean by 'universal reasoning abilities'? One could ask that an agent capable of 'universal reasoning' should not exhibit such unsafe behaviors.

W4: The analysis in Section 4.2 claiming the existence of 'dangerous neurons' is based on correlation, not causation. To support the claim, the authors should provide a causal analysis investigating the effect of ablating such neuron activations on safety, which is missing. In addition, as the neuronal activation can significantly vary depending on the context, I think the statistical significance of choosing dangerous neurons based on Eq.5 should be reported together.

W5: missing information in Fig.3 - which model size did you use?

W6: Minor formatting issues - I think that many parts in the paper (L129-130, L198-199, L212-233, L285-286) don't follow the spacing guidelines. I don't mark it as a major paper formatting concern, but this should be fixed in the version.

W7: L117 - given additional reasoning steps, LVLMs can recognize the correct their outputs. -> I couldn't understand this sentence. Maybe a typo?

W8: Eq. 1 is hard to comprehend, and some terms are not precisely defined: for example, the definition of $\bigoplus$ is unclear. Does it mean concatenation? I think Eq.1 does not clarify the idea but adds unnecessary complexity.

W9: The same notation $y_s$ is used both in Eq.1 and Eq.4, which have different meanings.

W10: L223-224 - $w_k$ is not precisely defined when this term is first introduced.

W11: Regarding Fig.8 - axis labels and metric definitions are missing both in the figure and the caption, making it hard to interpret.

---

> ### Author Rebuttal · Authors · 2025-07-31
>
> >Q1: Limited novelty
>
> **A1:**   We respectfully disagree with the reviewer’s characterization of our main contribution.
> 1. On the novelty and positioning relative to prior work (re: "Crescendo")
> Inspired by a series of works on cascade effects in LLMs, and what has been described as the safe-input–unsafe-output phenomenon, we already cite several relevant papers in L42, L43, and L62.
> These works—focusing on escalation-style failures and multi-turn jailbreaks —propose intuitions that are similar to Crescendo, and in fact many of them precede it in publication.’
> As such, Crescendo can be naturally positioned within this broader class of multi-turn jailbreak studies.
> In our revision, we will further strengthen this connection by explicitly including a broader class of multi-turn jailbreak studies in Line 43, and revising the phrasing as follows:“This phenomenon has been previously explored in text-to-image generation models, LLMs [1,2,3], and LVLMs..”
>
> [1] Russinovich, M., et al. (2025). The Crescendo multi-turn LLM jailbreak attack. In USENIX Security 2025.
>
> [2] Sun, X., Zhang, D., et al. (2024, August). Multi-Turn Context Jailbreak Attack on Large Language Models From First Principles. arXiv.
>
> [3] Zhou, Z., Xiang, J., et al. (2024, February). Speak Out of Turn: Safety Vulnerability of Large Language Models in Multi-turn Dialogue. arXiv.
>
>
>
> 2.  “the main contribution of this work is its expansion to various VLM safety domains.”
> We disagree with the interpretation that “the main contribution of this work is its expansion to various VLM safety domains.”
> As stated in Line 47, the core contribution of our work is to show that:
>
> “A wide range of images can potentially be exploited to jailbreak LVLMs and generate harmful outputs by combining additional safe images and prompts.”on contribution of our work’’
>
> This uncovers a generalizable failure mode in multimodal alignment, where inputs that appear benign can nonetheless induce harmful model behavior. We study this failure mode in depth throughout the paper. The Snowball strategy is not the main novelty, but rather a technique to enhance attack success rates (JSR) under our broader attack formulation.
>
> 3. ‘but the analysis and insights for the SSA's behavior depending on the domains are not thoroughly explored.’
>
> In Section 2, we introduce the safety snowball effect, a general vulnerability in LVLMs where initially ambiguous responses escalate into harmful ones. Using MM-SafetyBench, we explore this in multimodal settings with contextually safe (not explicitly harmful) inputs. In Section 3, we present SSA—a domain-agnostic agent that uses both visual and textual inputs to trigger and amplify unsafe behavior. Experiments in Section 4 show SSA works across image domains: Figure 6 shows even safe-looking images can cause jailbreaks, while Figures 7 and 8 reveal both safe and unsafe inputs activate the same dangerous neurons. These results demonstrate SSA’s consistent and robust behavior across modalities.
>
> >Q2: The subjectivity of 'not overly harmful prompts'.
>
> **A2:**  The analysis in Table 12 was a supplementary experiment, intended to support our claim that the initial prompts used in SSA are not overtly harmful. Since the reviewer raised concerns about category diversity, we conducted additional evaluations specifically to address this point.
> We expanded the experiment beyond the violence category to include another representative domain: self-harm. As shown in Table R1 below:
>
> **Table R1. Safety evaluation of SSA initial prompts in the Self-Harm category across different models and evaluator configurations.**
> | Model         | 1st LLaMA | 1st Guard | 1st OpenAI | 2nd LLaMA | 2nd Guard | 2nd OpenAI |
> | ------------- | --------- | --------- | ---------- | --------- | --------- | ---------- |
> | GPT           | 0.11      | 0.12      | 0.12       | 0.38        | 0.16      | 0.37       |
> | InternVL      | 0.11      | 0.12      | 0.13       | 0.42     | 0.28      | 0.35       |
> | LLaVAone      | 0.09      | 0.07      | 0.11       |  0.12      | 0.15      | 0.22     |
> | miniCPM       | 0.16      | 0.09      | 0.16       | 0.19      | 0.16      | 0.31       |
> | QwenVL2       | 0.17      | 0.11      | 0.14       | 0.24     | 0.26      | 0.25       |
> | VILA          | 0.11      | 0.05      | 0.19       | 0.21     | 0.20      | 0.33       |
>
> These results support our original finding: the initial prompts used in SSA are not inherently harmful across multiple sensitive domains, as confirmed by high safety scores and low flagged rates across evaluators.
>
>
> >Q3: 'Universal Reasoning Abilities' Not Defined
>
> **A3:**  We define universal reasoning abilities in L55 of the manuscript. This definition aligns with prior work where such reasoning capabilities are studied in both LLMs and LVLMs [1,2].
>
> Although it is intuitive to assume that stronger reasoning capabilities would correlate with safer behavior, in practice, the opposite is often observed—models with more advanced reasoning abilities tend to be more vulnerable to adversarial misuse and jailbreaks [3, 4].
>
> [1] Lee, S., Sim, W., Shin, D., Seo, W., Park, J., Lee, S., et al. (2024). Reasoning abilities of large language models: In-depth analysis on the abstraction and reasoning corpus. ACM Trans. Intell. Syst. Technol.
>
> [2] Li, Y., Liu, Z., Li, Z., Zhang, X., Xu, Z., Chen, X., et al. (2025). Perception, reason, think, and plan: A survey on large multimodal reasoning models. arXiv.
>
> [3] Zhou, K., Liu, C., Zhao, X., Jangam, S., Srinivasa, J., Liu, G., et al. (2025). The hidden risks of large reasoning models: A safety assessment of R1. arXiv.
>
> [4] Kuo, M., Zhang, J., Ding, A., Wang, Q., DiValentin, L., Bao, Y., et al. (2025). H-CoT: Hijacking the chain-of-thought safety reasoning mechanism to jailbreak large reasoning models, including OpenAI O1/O3, DeepSeek-R1, and Gemini 2.0 Flash Thinking. arXiv.
>
> >Q4: Causality and Correlation in 'Dangerous Neurons'
>
> **A4:**  The analysis in Section 4.2 follows a widely used neuron-level approach, similar to prior work on identifying safety-relevant or functionally interpretable neurons in LLMs and LVLMs [1,2,3]. These works, like ours, use activation differences between safe and unsafe outputs to surface neurons potentially associated with harmful behavior.
>
> We agree that establishing causality would be ideal. However, ablating neuron activations alone does not imply a causal relationship either, as emphasized in causal inference literature[4,5] . The activation of a single neuron cannot be treated as an isolated causal variable—it's part of a distributed computation, and modifying it can produce entangled or ambiguous effects downstream.
>
> [1] Chen, J., Wang, X., et al. (2024). Finding safety neurons in large language models. arXiv:2406.14144.
>
> [2] Hoang-Xuan, et al. (2024). LLM-assisted concept discovery: Automatically identifying and explaining neuron functions. arXiv:2406.08572.
>
> [3] Ahn, Y. H., et al. (2024). WWW: A unified framework for explaining what, where, and why of neural networks. In ICCV.
>
> [4] Pearl, J., & Mackenzie, D. (2018). The Book of Why: The New Science of Cause and Effect. Basic Books.
>
> [5] Bau, D., et al. (2020). Understanding the role of individual units in a deep neural network. PNAS.
>
>
> >Q5: Model Version in Fig. 3
>
> **A5:**  Thanks for your question. We used the Qwen‑VL2 72B variant in all experiments. This is mentioned in the "Models" subsection of Section 6 (Experiments). We agree that this information should be more prominent. We will revise the figure caption and clarify the model version again in the main text to ensure there is no confusion. Thank you for bringing this to our attention.
>
> >Q6: Formatting (L129-286)
>
> **A6:** Thanks for Pointing it out. We will correct all spacing to comply with NeurIPS guidelines.
>
> >Q7: L117 Typo
>
> **A7:** The sentence “LVLMs can recognize the correct their outputs” will be corrected to “LVLMs can recognize and correct their outputs when provided additional reasoning steps.”
>
> >Q8:  Eq. 1 Clarity
>
> **A8:** Thank you for the helpful feedback. We agree that the notation in Equation 1—especially the meaning of ⊕—was not clearly introduced. In the revised manuscript, we will explicitly define ⊕ as string concatenation and rephrase the sentence to make the construction of the prompt easier to follow.
>
>
> >Q9: Notation Overlap in Eq. 1 & Eq. 4
>
> **A9:**  Thank you for catching that. We use the same notation in both places, but in Eq.1 it refers to the overall two‑stage attack pipeline, wrapping the entire process (image generation, both prompts and the final evaluation) into a single mapping, whereas in the Eq.4  it denotes only the harmful‑snowballing phase of that pipeline—showing how the model builds on its initial unsafe response. We will clarify this distinction in the text to avoid any confusion from notation reuse. They have the same meanings.
>
> >Q10:  Undefined Notation at L223
>
> **A10:** Thank you for pointing this out. We agree that the notation introduced at L223–224 could be confusing without a proper definition. In the revised manuscript, we will define each symbol upon first use. The updated version will clarify the input structure and the meaning of the response variables.
>
> Before (original):
> For a given input w = ⟨v, w₀, ..., wₜ⟩, which includes an image v and a question, we denote ...
>
> After (revised):
> ..., where v denotes the input image and w₀ to wₜ are the initial textual tokens (e.g., a question), we define the unsafe jailbreak response as wᵤ = ⟨wₜ₊₁, ..., wₜ₊ₘ⟩, and the safe response as wₛ = ⟨w′ₜ₊₁, ..., w′ₜ₊ₙ⟩.
>
> >Q11: Fig. 8 Axis Labels
>
> **A11:**  As mentioned in the current caption, each grid in Figure 8 represents a single neuron. We agree that adding axis labels—such as neuron indices and activation differences—would improve clarity. In the revised version, we will update both the figure and caption to explicitly include these axis labels and metric definitions to make the visualization easier to interpret.

---

> > ### Comment · Reviewer_td69 · 2025-08-03
> >
> > Thank you for providing a thorough and clear response. After reading the response, I recognize that the main contribution of this work lies in revealing that safe images can be used to trigger unsafe outputs, which is not known in previous VLM jailbreaking literature. In addition, the table provided by the authors resolves W2, and I believe the paper's clarity of presentation (which was another major concern) can be greatly improved in the revised version, as the authors have well addressed A5-A11.
> >
> > If time permits, I would like to ask several follow-up questions to make sure I understand this work clearly:
> >
> > 1. Regarding A3, I see the connection between universal reasoning abilities and overinterpretation as stated in L55. However, this overinterpretation behavior is only discussed qualitatively, and I think it remains unclear what overinterpretation is, how it can be measured, and which aspects of the model's generated explanation (intended to be safe) are classified as 'malignant overinterpretation' that make the model vulnerable in the next turn of the conversation. It somewhat connects to my earlier question (in A1) regarding the analysis of domain-dependent behaviors, because I think we could gain further insights on this by observing the similarities and differences in the model's response patterns or attack successes depending on different domains. Could you share your thoughts on this point?
> >
> > 2. On dangerous neurons (A4), I agree that an ablation-based study alone does not ensure causality, as it might be a part of the distributed computation. However, with the same argument, I believe naming the model components that showed high activation patterns when generating unsafe output as 'dangerous neurons' from the observed correlations can be misleading (as such activation patterns can also play other roles, or they might not necessarily play a significant role in modulating safety). I believe there is room for a more thorough analysis, including the ablation study as the most straightforward approach to gain further insights. Without them, I'd recommend tempering the tone or consider renaming the term 'dangerous neurons'.

---

> > > ### Author Response · Authors · 2025-08-04
> > >
> > > Thank you for your follow-up—it was sharp, thoughtful, and helped us see several parts of our work more clearly. We're glad you recognized our central point: that even ordinary or "safe" images can elicit unsafe outputs from large vision-language models. This, we think, highlights a subtle but important failure mode in multimodal alignment that deserves more attention.
> > >
> > > We also appreciate that our clarifications addressed your earlier concerns around prompt design and result presentation (W2, A5–A11). Your comments encouraged us to revisit parts of the framing, and we’ve revised several areas with more care as a result.
> > >
> > > Regarding your question on *universal reasoning abilities* and *overinterpretation*, we agree that our current discussion remains qualitative and would benefit from greater formalization. You're right to point out that this connects to domain-specific behaviors. Although we initially introduced this case as an example of the model's reasoning tendencies, we acknowledge that the precise mechanisms driving such malignant overinterpretation are not yet well understood. There, a seemingly safe image of a bank interior—labeled “VERY\_UNLIKELY” to be harmful by an external vision safety classifier—when paired with a neutral prompt, leads GPT-4o to generate content describing the planning stages of a bank robbery. This emergent behavior arises not from explicitly harmful inputs, but from the model’s tendency to infer unintended causal or thematic links between visual and textual elements.
> > >
> > > Based on your feedback, we will make this connection more explicit in the revised paper. In Sections 1 and 2, we will define overinterpretation more precisely as the model drawing unsafe or aggressive associations between otherwise neutral elements, likely due to strong cross-modal generalization. We agree that studying how these patterns vary across domains—and tracking activation paths—could offer valuable insight into this broader issue.
> > >
> > > On the term dangerous neurons, we appreciate your perspective and agree that the current naming may inadvertently overstate the case, especially given that our analysis is correlation-based.  As you suggested, we'll use terms like high-risk or safety-associated neurons instead, unless future work shows a stronger causal link. We're currently testing this with activation patching and sparse ablation methods. If the results are ready in time, we’ll include them in the updated version to add more rigor.
> > >
> > > Thanks again for your close reading and direct questions. Your feedback improved both our thinking and our writing, and we’ve incorporated it throughout the new draft.

---

> > > > ### Comment · Reviewer_td69 · 2025-08-06
> > > >
> > > > Thank your further clarifications. Based on the discussion, I will raise the score by 1, as my major concern regarding the contribution of this work and clarity has been resolved, and this work reveals a new way of VLM jailbreaking using safe images. That said, I think many aspects of this work can be much improved to understand the underlying mechanisms (e.g., detailed analysis of safety-related components and a more precise connection to the overexplanation behavior and safety issue, although the authors provided a hypothetical explanation), and the technical aspect of the jailbreaking method was straightforward given previous approaches.

---

### Official Review · Reviewer_5D2j · 2025-07-03

**Clarity:** 3
**Significance:** 3
**Originality:** 3
**Rating:** 4
**Confidence:** 3

**Summary:**

This paper introduces a novel framework for jailbreaking Large Vision Language Models (LVLMs), Safety Snowball Agent (SSA), using only seemingly safe inputs. SSA exploits two intrinsic properties of LVLMs universal reasoning ability and the safety snowball effect to escalate benign inputs into harmful outputs, unlike prior methods that relied mainly on harmful images or prompts. SSA operates in two stages: (1) Initial Response Generation, where a combination of safe images and subtly intentioned prompts are used to generate an ambiguous or weakly unsafe output, and (2) Harmful Snowballing, where iterative follow-up prompts guide the model into generating increasingly harmful content. The authors construct a new benchmark, SafeAttack-Bench, comprising images verified as safe, and demonstrate that SSA achieves state-of-the-art attack success rates on leading LVLMs. Furthermore, their analysis reveals that both safe and unsafe images activate similar dangerous neuron patterns, highlighting a fundamental safety vulnerability in LVLM architectures.

**Questions:**

- Is there any specific reason for the number of samples generated, 1347?
- Have the authors checked on images from [1], which has data from various categories?

[1] Luo, W., Ma, S., Liu, X., Guo, X., & Xiao, C. (2024). JailBreakV: A Benchmark for Assessing the Robustness of MultiModal Large Language Models against Jailbreak Attacks. arXiv preprint arXiv:2404.03027.

**Ethical Concerns:**

["NO or VERY MINOR ethics concerns only"]

**Final Justification:**

The authors have addressed my initial concerns satisfactorily. The paper is technically sound and the contributions are clearly presented. While the method is very novel and interesting, I believe there is high dependency on the existing benchmark MM-SafetyBench, which limits our understanding of the method without relying on it. It also limits the size to a comparatively smaller number of samples. Hence, I believe my rating fairly reflects its current impact and scope; therefore, I am maintaining my rating.

**Limitations:**

yes

**Quality:**

3

**Strengths And Weaknesses:**

**Strengths**
- The paper introduces a novel and significant vulnerability in LVLMs, the ability to jailbreak models using safe inputs. This shifts the focus from solely harmful inputs to understanding how benign combinations can lead to unsafe outputs, presenting an understanding of AI safety.
- The paper provides a solid motivation identifying the underlying mechanisms of this vulnerability, reasoning abilities, and the "safety snowball effect" as compelling reasons.
- The paper is well-written and clearly explains the proposed SSA framework and its two stages. The methodology for jailbreak image generation and harmful snowballing is detailed.

**Weaknesses**
- While the paper highlights a critical vulnerability, it offers limited discussion of potential mitigation strategies, given the significance of the discovered "safety snowball effect".
- The activation of dangerous neurons by safe inputs, a more in-depth discussion on how these insights could inform future defense mechanisms, would strengthen the paper.
- The method heavily relies on intent recognition. If the model is unable to find harmful intent or refuses to provide dangerous actions, this limits further generation.

---

> ### Author Rebuttal · Authors · 2025-07-31
>
> >Q1:The activation of dangerous neurons by safe inputs, a more in-depth discussion on how these insights could inform future defense mechanisms, would strengthen the paper.
>
> **A1:** Thank you for your valuable comment. We appreciate the reviewer’s insight regarding the activation of dangerous neurons by seemingly safe inputs and agree that a deeper discussion on how this phenomenon could inform future defense mechanisms would enrich the paper. In the revised version, we have expanded the discussion section to emphasize the implications of our findings: specifically, the fact that safe images can elicit neuron activations nearly identical to those triggered by unsafe content suggests that current safety filters, which often rely on input-surface cues, may be insufficient. This insight points to the need for more robust safety mechanisms that incorporate internal model interpretability—for example, by monitoring the activation patterns of identified “dangerous” neurons as early warning signals or incorporating neuron-level gating strategies to suppress harmful generation pathways. We believe these directions could offer promising foundations for developing next-generation safety solutions in LVLMs, and we are grateful for the suggestion that helped us refine this perspective.
>
>
> >Q2: The method heavily relies on intent recognition. If the model is unable to find harmful intent or refuses to provide dangerous actions, this limits further generation.
>
> **A2:** Thank you for this insightful observation. To assess whether this reliance imposes practical limitations, we conducted an evaluation of the model's ability to identify harmful intent on the MM-SafetyBench dataset. Specifically, we measured the intent recognition success rate—that is, the proportion of cases where the assistant model successfully recognized harmful topics without refusal.
> As shown in the table below, the expert model we use (GPT-4o) demonstrates a consistently high success rate across a range of harmful categories, indicating that intent recognition is rarely a bottleneck in practice:These results suggest that the current model is highly effective in identifying harmful intents, enabling our framework to proceed with subsequent stages of jailbreak generation in nearly all cases. We have clarified this point in the revised manuscript to address the reviewer’s concern. Thank you again for helping us strengthen the presentation of our method’s robustness.
>
> **Table R1. Intent recognition success rates of GPT-4o on MM-SafetyBench across different harmful categories.**
> | Violence | Self-Harm | Dangerous Object | Terrorism | Hate |
> |----------|-----------|------------------|-----------|------|
> |   0.95   |   0.97    |       0.95       |   0.96    | 0.97 |
>
> >Q3: Is there any specific reason for the number of samples generated, 1347?
>
> **A3:** Thanks for your question. We curated SafeAttack‑Bench by filtering all images labeled “VERY_UNLIKELY” for unsafe content by Google Cloud Vision Moderators. This process yielded precisely 1,347 qualifying images. We will clarify this selection procedure in Section 5.
>
> >Q4: Have the authors checked on images from [1], which has data from various categories?
>
> **A4:**  Thank you for the helpful suggestion. We have carefully reviewed the JailBreakV benchmark [1] and will include a discussion and citation in our revised manuscript. After examining the dataset, we found that the image categories in JailBreakV substantially overlap with those used in MM‑SafetyBench and our curated benchmark, particularly in terms of coverage of safety‑critical scenarios—including illegal activity, hate speech, and privacy violence. Additionally, several of the attack strategies used in JailBreakV—such as figstep and template-based prompting—are already included in our baseline methods for MMSafetyBench. Based on this alignment, we believe our experimental results on MMSafetyBench are representative and comparable to those that might be obtained on JailBreakV.
>
>
> [1] Luo, W., Ma, S., Liu, X., Guo, X., & Xiao, C. (2024). JailBreakV: A Benchmark for Assessing the Robustness of MultiModal Large Language Models against Jailbreak Attacks. arXiv preprint arXiv:2404.03027.

---

> > ### Comment · Area_Chair_jJcC · 2025-08-06
> >
> > Dear Reviewer,
> >
> > A gentle reminder that the extended Author–Reviewer Discussion phase ends on August 8 (AoE). According to this year's new NeurIPS policy, Reviewers must participate in discussions with authors before submitting the “Mandatory Acknowledgement”.
> >
> > Please read the authors’ rebuttal and participate in the discussion ASAP. If your concerns have been resolved, please acknowledge this clearly. If not, please communicate what remains unaddressed.
> >
> > Thank you for your timely and thoughtful contributions.
> >
> > Best regards,
> >
> > AC

---

> > ### Comment · Reviewer_5D2j · 2025-08-06
> > **Official Comment by Reviewer 5D2j**
> >
> > I thank the authors for their thorough rebuttal. They have addressed my initial concerns satisfactorily, and I appreciate the clarifications provided.
> > That said, while the paper is technically sound and the contributions are clearly presented, I believe my rating fairly reflects its current impact and scope, therefore, I am maintaining my rating.

---

### Official Review · Reviewer_Akgd · 2025-07-03

**Clarity:** 3
**Significance:** 2
**Originality:** 3
**Rating:** 5
**Confidence:** 4

**Summary:**

This paper describes a new kind of LVLM vulnerabilitiy: when combined with carefully constructed images and prompts, seemingly safe images can still lead to generating harmful content. The authors identify two key properties that enable this phenomenon: universal reasoning capabilities (leading to overinterpretation of relationships across safe inputs) and a safety snowball effect (an initial unsafe response cascades into increasingly harmful outputs). Based on these, the authors propose the Safety Snowball Agent (SSA), an automated jailbreak framework that operates in two stages. In the first stage, SSA retrieves jailbreak images to generate responses. In the harmful snowballing stage, SSA refines prompts in a way that leads to increasingly harmful generations. The evals span 7 LVLMs, and focus on an existing benchmark (MM-SafetyBench) and a new dataset, SafeAttack-Bench. The findings demonstrate high attack success rates.

**Questions:**

1. Attack Generalization. The paper claims "any image" can be exploited (sec 4, line 78), but the evaluation uses curated datasets. Have the authors conducted experiments with successful attacks using completely random, unrelated images—perhaps even outside the domain of natural images (pictures one could capture)?
2. Eval Methodology. Heavy reliance on gpt-4o for evals raises some concerns about eval bias. How sensitive are the results to the choice of evaluator, and have the authors validated against human evaluation? I realize that a minor human review was employed but that does not appear to be very systematic and reproducible from the current description.
3. Baseline Strength. The baseline methods in tab 1 show surprisingly low performance — are these baselines implemented correctly, and do they actually represent the current SOTA in LVLM jailbreaking?

**Ethical Concerns:**

["NO or VERY MINOR ethics concerns only"]

**Final Justification:**

All of my concerns have been addressed in a satisfactory manner, and so I see no reason not to move my "Borderline Accept" to an "Accept".

**Limitations:**

The authors adequately address limitations in app A.8.4, though, it seems appropriate to move this up to the main body. The broader impact discussion (A.8.5) appropriately highlights risks and encourages responsible AI development. However, the paper could benefit from more concrete recommendations than high-level proclamation.

**Paper Formatting Concerns:**

Overall, the adherence to the formatting instructions seems good, though I would recommend the authors to invest in figure legibility and overall quality could be improved (font size in figs 6-7 and the activation figs 8-9).

**Quality:**

3

**Strengths And Weaknesses:**

Strengths:

1. Novel & Well-Motivated Framing. This paper represents a significant departure from existing jailbreak research that focuses primarily on harmful inputs, which is refreshing.
2. Comprehenisve Experiments. Seven recent LVLMs are used for evals with multiple evaluation metrics. The statistical rigor is enhanced by inclusion of confidence intervals, t-statistics, and p-values.
3. Thorough Analysis of Underlying Mechanism. Compelling evidence is presented in the form of neuron activation analysis (sec 4.2, figs 7-8), showing that safe and unsafe images activate nearly identical dangerous neurons.

Weaknesses:

1 Limited Defense Evaluation. The paper lacks exploration of more sophisticated defense mechanisms or mitigation strategies.
2. Dependency on External Tools. The SSA framework relies on external services (Google Custom Search API, Google Cloud Vision, GPT-4o) for image generation and evals (sec 3.3, app A.4), which may limit reproducibility and generalizability.
3. Unclear Baseline Comparisons. SSA wins on all the benchmarks but the reported baselines seem to be performing very poorly across the boards — I wonder if those are good baseline to begin with and whether better methods exist that this paper could compare against.

---

> ### Author Rebuttal · Authors · 2025-07-31
>
> >Q1:Attack Generalization. The paper claims "any image" can be exploited (sec 4, line 78), but the evaluation uses curated datasets. Have the authors conducted experiments with successful attacks using completely random, unrelated images—perhaps even outside the domain of natural images (pictures one could capture)?
>
> **A1** Thank you for raising this important point about generalization.
>
> To test whether completely unrelated or out-of-domain images can trigger unsafe behavior, we conducted additional experiments with **25 pure color images** and **25 Random noise images**, which contain no semantic content or recognizable structure.
> The results are summarized  in Table R1.
>
> **Table R1. Generalization results on out-of-domain images. “Noise” = random noise; “Pure” = solid color. Higher values indicate more unsafe responses.**
>
> | Image Type | Unsafe Ratio | GPT Score ↑ | OpenAI Moderation ↑ |
> | ---------- | ------------ | ----------- | ------------------- |
> | **Noise**  | 0.720        | 2.96        | 0.317               |
> | **Pure**   | 0.045        | 1.05        | 0.078               |
>
> These results suggest that even non-natural images, particularly those with high-frequency patterns like noise, can reliably induce unsafe generations under our attack framework. In contrast, pure color images—which are visually uniform and featureless—tend to fail in eliciting unsafe behavior.
>
> One possible explanation is that noise images, despite being semantically meaningless, still activate diverse visual pathways within the LVLM due to their texture-like statistical properties. These activations may inadvertently align with patterns seen during multimodal pretraining, making the model more susceptible to exploitation. Pure color images, by contrast, may not produce sufficiently rich visual activations to trigger model behavior.
>
> Our main benchmark already includes a subset of **non-natural images** (e.g., abstract art, schematic diagrams), and we observed similar vulnerabilities in those settings. These findings reinforce our broader claim that **a wide range of images** has the *potential* to be exploited for jailbreaking in LVLMs—though the degree of effectiveness varies.
>
>
>
>
>
>
>
>
> >Q2: Eval Methodology. Heavy reliance on gpt-4o for evals raises some concerns about eval bias. How sensitive are the results to the choice of evaluator, and have the authors validated against human evaluation? I realize that a minor human review was employed but that does not appear to be very systematic and reproducible from the current description.
>
> **A2.** Thank you for raising this important point. We fully recognize the concerns around potential bias when relying on GPT-4o for evaluation. In addition to GPT-4o, we also employed OpenAI's content moderator and LLaMA Guard to provide complementary perspectives on evaluation quality and safety.
>
> While our human evaluation was limited in scale, we aimed to make it as systematic and reproducible as possible (see Appendix A.5.3). Specifically, we conducted a structured manual review using standardized questionnaires to compare GPT-4o’s automatic assessments with human judgments. Given the substantial cost and effort involved in scaling high-quality human evaluation, we see this as a valuable but resource-intensive complement to model-based evaluation.
> To further examine the sensitivity of results to the choice of evaluator, we conducted an additional analysis by replacing GPT-4o with Gemini as the automatic evaluator and comparing the resulting harmfulness scores across different unsafe content categories. The comparison showed that while overall trends remained consistent.
> In the Table R2, we present the differences in harmfulness scores when using Gemini-1.5-pro instead of GPT-4o as the evaluator (i.e., Gemini score – GPT score):
>
> **Table R2. Differences in harmfulness scores when replacing GPT-4o with Gemini-1.5-pro as the evaluator (Gemini − GPT). Positive values indicate higher harmfulness judged by Gemini.**
>  | Model           | Celebrity | Violence | Self-harm | Objects | Average |
> |----------------|-----------|----------|-----------|---------|---------|
> | Intern-VL      | +0.11      | +0.07    | -0.11     | +0.09   | +0.09   |
> | + Initial Resp | -0.15     | +0.17    | +0.14     | -0.08   | -0.06   |
> | + SSA          | -0.15     | -0.15    | +0.11     | -0.16   | -0.13   |
> | VILA           | +0.08      | +0.07    | -0.12     | -0.07   | -0.08   |
> | + Initial Resp | -0.15     | +0.08    | +0.07     | -0.07   | +0.07   |
> | + SSA          | -0.05     | -0.16    | +0.14     | -0.11   | -0.35   |
> | Qwen-VL2       | -0.13     | +0.07    | -0.11     | +0.10   | +0.05   |
> | + Initial Resp | +0.07     | +0.14    | +0.13     | +0.03   | -0.05   |
> | + SSA          | +0.07     | -0.13    | +0.05     | -0.06   | -0.11   |
> | GPT-4o         | +0.07     | +0.08    | +0.07     | +0.08   | -0.08   |
> | + Initial Resp | +0.08     | -0.12    | -0.05     | +0.06   | -0.11   |
> | + SSA          | +0.15     | +0.05    | +0.07     | +0.06   | +0.04   |
>
>
> >Q3: Baseline Strength. The baseline methods in tab 1 show surprisingly low performance — are these baselines implemented correctly, and do they actually represent the current SOTA in LVLM jailbreaking?
>
>
> **A3:** Thank you for the question. The baseline methods in Table 1 were implemented following existing jailbreaking strategies (e.g., prompt fuzzing, typos, and formatting tricks). However, their low performance reflects a trend: many modern LVLMs—especially those from OpenAI, Meta, and Qwen—have incorporated strong safety alignment mechanisms that effectively block these shallow attacks. For example, techniques like injecting typos into visual text were once effective, but are now easily detected or ignored by well-aligned models. Models like VILA, which lack such alignment, remain vulnerable to these basic methods—showing that the attack space remains relevant but increasingly model-dependent. We'll clarify this in the revised text to avoid the impression that these baselines were misapplied.

---

> > ### Comment · Reviewer_Akgd · 2025-08-06
> >
> > Thank you for taking the time to respond to all of my points. All of the responses seem sound to me, and so I'm happy to move the 'Borderline Accept' to an 'Accept'.

---

### Note · Authors · 2025-08-16

---

**Dear ACs/SACs/PCs,**

We would like to summarize the strengths of this work acknowledged by the reviewers, and our responses to address all the
 reviewers' concerns.

---

### **Strengths highlighted by reviewers**

1. **Novel vulnerability surfaced** – Safe multimodal inputs can escalate into harmful outputs via a safety snowball effect (Akgd, 5D2j).
2. **SSA design** – Two-stage, automated framework exposing this failure mode across diverse LVLMs (5D2j, fnZy).
3. **Broad, rigorous evaluation** – Seven LVLMs with multiple metrics, CIs, statistical tests (Akgd).
4. **Neuron-level analysis** – Safe and unsafe images activate nearly identical safety-associated neurons (Akgd, td69).
5. **Readability** – Paper well-written, easy to follow, with solid motivation (Akgd, fnZy).

---

### **Main concerns raised**

1. **Generalization (Akgd)** – Does SSA work on “non-natural” images (noise, pure color)?
2. **Evaluator sensitivity & human checks (Akgd)** – Heavy reliance on GPT-4o may bias results; need cross-checks and human validation.
3. **Baselines & fidelity + failure modes (Akgd, fnZy)** – Are baselines too weak? Under what conditions does SSA fail?
4. **Clarity (5D2j, td69)** – Benchmark details, terminology like “universal reasoning abilities,” “dangerous neurons.”

---

### Our **responses** to these concerns:

1. **Generalization** – **Conducted new experiments** on 25 random-noise and 25 pure-color images. Confirms exploitability beyond natural images.
2. **Evaluator sensitivity & human checks** – **Re-ran evals with Gemini-1.5-pro instead of GPT-4o**, trends remained stable. **Structured human review added**, with plan to report cross-evaluator agreement.
3. **Baselines & fidelity + failure modes** – **Clarified baselines followed prior strategies**; weak results reflect strong modern alignment. **Documented SSA limitations**, e.g., fails when initial prompts include highly flagged keywords (e.g., “bomb,” “child pornography”).
4. **Clarity** –  We revised tables, captions, and figures for clearer comparisons to improve overall clarity and reproducibility.
---


All major concerns have been addressed with **new experiments, new discussion, clarification of baselines, and refined terminology**. We thank the reviewers and ACs for their constructive feedback and hope these improvements are considered.

Sincerely,

**Authors**

---

---

### Decision · Program_Chairs · 2025-09-17

**Decision:**

Accept (poster)

**Comment:**

The paper studies the safety of Large Vision-Language Models (LVLMs) and reveals that a safe image can be exploited to achieve jailbreaks when combined with other safe images and prompts. Building on this observation, the authors propose the Safety Snowball Agent (SSA) to jailbreak LVLMs. The experiments cover 7 LVLMs and two categories of red-teaming benchmarks.

Strengths:

- Introduces a novel and important vulnerability in LVLMs: the possibility of jailbreaking models using only safe inputs.

- Provides extensive evaluation across 7 recent LVLMs, demonstrating the generality of the phenomenon.

Weaknesses:

- The technical novelty of the proposed jailbreaking method is limited.

- The defense evaluation is limited.

During the rebuttal, the authors provided additional explanations and experimental results that addressed most of the concerns raised in the reviews. Overall, the paper highlights an underexplored and practically important vulnerability in LVLMs. Despite limitations in technical novelty, the insights and empirical evidence make the work valuable to the community.